# Contrasting Behavioral and Electrophysiological Responses of *Eucryptorrhynchus scrobiculatus* and *E. brandti* (Coleoptera: Curculionidae) to Volatiles Emitted from the Tree of Heaven, *Ailanthus altissima*

**DOI:** 10.3390/insects12010068

**Published:** 2021-01-14

**Authors:** Xiaojian Wen, Kailang Yang, Jaime C. Piñero, Junbao Wen

**Affiliations:** 1Laboratory of Forest Pathogen Integrated Biology, Research institute of Forestry New Technology, Chinese Academy of Forestry, Beijing 100091, China; wenxj1016@126.com; 2College of Forestry, Northwest A&F University, Yangling 712100, China; yangkl@nwafu.edu.cn; 3Beijing Key Laboratory for Forest Pests Control, College of Forestry, Beijing Forestry University, No. 35, Tsinghua East Road, Beijing 100083, China; 4Stockbridge School of Agriculture, University of Massachusetts, Amherst, MA 01003, USA; jpinero@umass.edu

**Keywords:** host plant volatiles, semiochemical, feeding selection, behavior, physiology

## Abstract

**Simple Summary:**

*Eucryptorrhynchus scrobiculatus* (Motschulsky) and *E. brandti* (Harold) are significant pests of tree of heaven, *Ailanthus altissima*, often leading to tree death. Monitoring systems that involve host–plant based attractants need to be developed for both insect pest species. Here, we compared the behavioral and electrophysiological responses of *E. scrobiculatus* and *E. brandti* to volatiles emitted by various parts of the host plant. Host plant-derived volatiles showed to play a greater role in the foraging behavior of *E. brandti* than in *E. scrobiculatus*. Volatile components of phloem were found to be particularly attractive to *E. brandti*.

**Abstract:**

*Eucryptorrhynchus scrobiculatus* and *E. brandti* (Coleoptera: Curculionidae) are host-specific pests of *Ailanthus altissima* (Mill.) Swingle (Sapindales: Simaroubaceae), causing extensive damage to the host. There are no effective attractants available for pest management. The main aim of this study was to explore the role of host plant-derived volatiles in the behavioral response of both weevil species. In a field experiment, both weevil species showed positive response to phloem, and there was no preference for phloem associated with healthy or injured trees. Significantly more *E. brandti* adults responded to the olfactory treatments compared to *E. scrobiculatus*. In a large-arena experiment, both males and females of *E. scrobiculatus* significantly preferred phloem from the tree trunk while adults of *E. brandti* responded in significantly greater numbers to tree limbs than to any other parts of host. Females and males of *E. scrobiculatus* responded positively to all parts of host tested in the Y-tube bioassay, while *E. brandti* adults were only attracted by the phloem from healthy and injured trees. There were dissimilar electroantennographic responses to compounds such as 1-hexanol and (1*S*)-(−)-*β*-pinene between the two weevil species. This study represents the first report documenting behavioral and electrophysiological responses of *E. scrobiculatus* and *E. brandti* to volatiles from various parts of *A. altissima* and findings may aid efforts to develop attractants.

## 1. Introduction

Plant volatiles are known to play an important role in the host–location process of many species of insect herbivores [1]. Generally, it is believed that species with overlapping habitats have similar ecological niches and thus compete for food resources. Potential limitation in the availability of food resources can lead to differences in the abilities of competing species to exploit resources, often leading to dominance of one species over another [2,3]. However, if two species differ slightly in their perception of host plants and ensuing foraging behavior, that may not be the case. Such subtle behavioral or sensory differences might facilitate subtle niche shifts and allow species with similar niches to coexist [4,5].

*Eucryptorrhynchus scrobiculatus* (Motschulsky) and *E. brandti* (Harold) (Coleoptera: Curculionidae) are two closely related species of insect herbivores that feed on the tree of heaven, *Ailanthus altissima* (Mill.) Swingle (Sapindales: Simaroubaceae). Both weevil species are host-specific and adults and larvae feed on different parts of *A. altissima* [6,7,8]. For instance, *E. scrobiculatus* adults feed on annual branches, perennial branches, and petioles, whereas the larvae feed on the root system. In contrast, *E. brandti* adults feed on the trunk, and larvae develop under the bark, destroying the phloem and xylem as they feed [9,10,11,12]. The simultaneous occurrence of the two weevils causes extensive damage to *A. altissima*, commonly leading to tree death [10,13,14].

Developing effective pest management methods for *E. scrobiculatus* and *E. brandti* involving the application of insecticides [15,16] and novel trapping systems [17,18] has been the focus of recent research. In terms of trapping, previously, Yang et al. [17] found that a mixture composed of vinegar, ethanol, and apple juice was attractive to *E. scrobiculatus*. However, this attractant was not deemed feasible for mass production because it was made of fresh materials, resulting in a short span of attractiveness. Recently, trunk trap nets for *E. scrobiculatus* and adhesive trunk trap nets for *E. brandti* were developed [14,17,18,19]. Trunk trap nets would be more effective if attractants were made available. In order to develop effective attractants for both *E. scrobiculatus* and *E. brandti*, it is important to understand the role that volatiles of *A. altissima* play on the olfactory response of both weevil species. Previous observations (Wen, unpub. data) indicated that *E. scrobiculatus* and *E. brandti* congregate on the bare phloem of tree of heaven in the field, regardless of whether the trees are healthy or injured. Based on these observations, we inferred that volatiles from bare phloem may be attractive to the weevils.

The main goal of this study was to explore the role of host volatiles in the foraging behavior of *E. scrobiculatus* and *E. brandti*. In particular, we compared the effects of bare phloem from healthy and injured trees and volatiles emitted by different parts of host plant on the behavioral response of the weevils. In addition, volatile compounds emitted by different parts of *A. altissima* were collected by two methods (headspace solid phase microextraction (HS-SPME) and dynamic headspace method) and analyzed by gas chromatography-mass spectrometry (GC-MS). Finally, the antennal responses of *E. scrobiculatus* and *E. brandti* to selected chemical compounds were examined by electroantennography (EAG) and behaviorally using Y-tube olfactometer. These results may provide a reference for the development of attractants that are based on host–plant-derived volatiles.

## 2. Materials and Methods

### 2.1. Attractiveness of Bare Phloem of Healthy and Injured Trees to Adult E. scrobiculatus and E. brandti

The experiment was conducted at Xiaoxingdun village, Pingluo County, Ningxia Hui Autonomous Region (38°51′ 24″ N, 106°31′38″ E) in August 2017. *Ailanthus altissima* trees (tree height was about 10 m; diameter at breast height was about 30 cm; the inter-tree distance was 3.5 m) were planted as windbreak near farmland, and they received no fertilizers or pesticides. For this study, we used five trees that had healthy foliage and no signs of weevil infestation and five trees that were seriously infected with both weevil species (i.e., tree branches with some level of defoliation and presence of >50 emergence holes on the trunk). Trees were randomly selected.

We quantified the response of both weevil species using a mark-release-recapture method. For each weevil species, previously collected in the field, adults were starved for 12 h and color-marked with blue oil paint (Jing Dian brand, Beijing Sheng Shi Jing Dian Coating Technology Co., Ltd., Beijing, China) on the elytra before the releases [19]. At 06:00, a square (8 × 8 cm^2^) was cut from the trunk at breast height (1.5 m), and the bark was removed, thereby exposing the xylem and phloem. At 07:00, we released a group of 20 marked *E. scrobiculatus* and 20 marked *E. brandti* (mixed sexes) on the ground, 1 m away from each tree. The number of marked weevils arriving to the square area was counted every hour from 08:00 to 19:00 for 48 h. Each weevil was removed and collected in a container after counting.

### 2.2. Attractiveness of Various Parts of A. altissima to Adult E. scrobiculatus and E. brandti

The two experiments described below were conducted under laboratory conditions at Forestry Bureau in Lingwu city.

#### 2.2.1. Large Still-Air Arena Experiment 

This test quantified the response of adults of each species to four olfactory treatments: (1) annual branches (150 g), (2) freshly cut foliage from seedlings (height: 1 m; weight of plant material used: 150 g), (3) one tree limb (length: 10 cm; diam.: 7 cm), and (4) phloem (150 g) from the trunk of healthy *A. altissima* trees. These four parts of host plant were collected in the morning of a test day and individually placed on white paper plates (diam.: 20 cm). An empty plate was used as a control. These five treatments were arranged in a pentagonal shape. The distance between two adjacent treatments was 50 cm. All evaluations took place in the laboratory (28 °C, 60% r.h.), inside a screen cage (2 × 1.8 × 1.7 m^3^) constructed of mosquito netting made of polyester fabric.

For this test, 480 *E. scrobiculatus* and 480 *E. brandti* were field-collected and separated in groups of 20 weevils (10 *E. scrobiculatus* and 10 *E. brandti*) each. All weevils were starved for 12 h for the observations. Ten *E. scrobiculatus* and 10 *E. brandti* adults were then released in the center of the experimental arena at 08:00, and the number of weevils arriving to each treatment was recorded every 2 h from 10:00 to 20:00. Responding weevils were removed at each time interval. At the end of the trial, the experimental arena was cleaned up and left unoccupied overnight. For each trial, fresh plant material and new weevils were used and the position of the olfactory treatments was determined randomly. Trials were repeated 8 times, over a 15-day period.

#### 2.2.2. Y-Tube Bioassay

The olfactory response of *E. scrobiculatus* and *E. brandti* to different parts of *A. altissima* was quantified in a Y-tube olfactometer. The olfactory treatments evaluated were (1) phloem (20 g) from the trunk of healthy trees, (2) phloem (20 g) from the trunk of highly infested trees (>50 emergence holes on the trunk), (3) freshly cut foliage from seedlings (20 g), and (4) pieces of a randomly selected annual branch (20 g).

*Insects*. Adult weevils used for this experiment were starved for 24 h for *E. scrobiculatus* and 40 h for *E. brandti* before the bioassay. The length of starvation was chosen based on preliminary observations indicating that when adult *E. brandti* were starved for 24 h most of them stayed in the base tube and kept still during the test, while similarly-starved adults of *E. scrobiculatus* readily responded to the stimuli.

*Bioassays*. The Y-tube olfactometer consisted of a 15 cm base tube (diam. = 2 cm) with two 12 cm arms (diam. = 1.5 cm) connected at a 75° angle to two glass spherical traps (diam. = 6 cm) and glass conical flasks (V = 500 mL) that contained the plant material. Moistened activated charcoal-filtered air was pumped by an atmospheric sampler (QC-1S, Beijing Municipal Institute of Labour Protection, Beijing, China) into each of two flasks at a rate of 250 mL/min. Airflow rate was calibrated using an electronic flow meter. All observations were conducted between 09:00 and 18:00.

At the onset of the bioassays, 20 g of a particular plant material was placed inside one of the two conical flasks that were connected to the two arms of the Y-tube olfactometer. The second flask was empty and was used as a control. A score line was drawn on each of the two arms of the Y-tube that were associated with either a particular odor treatment or the control arm, at 3 cm from the intersection. Subsequently, an individual weevil was released at the entrance of the common arm of the Y-tube using a glass vial and the behavioral response was recorded for 5 min. The behavior of the test weevil was classified as “no-choice” if the insect remained in the base tube of the Y-tube olfactometer by the end of the observation period. The response was scored as “choice” if the weevil entered one of the two arms of the Y-tube, crossed the score line, and remained there for at least 1 min [20]. The position of the conical flasks with odor source in relation to each arm was reversed after each test, and all tested weevils were used only once. Eighteen males and eighteen females of *E. scrobiculatus* and twenty males and 20 females of *E. brandti* were tested for each treatment. We used a randomized block design with blocking over time. All glass devices were cleaned after each replication by rinsing with anhydrous ethanol and distilled water and then oven-dried.

### 2.3. Characterization of Volatiles from Different Parts of A. altissima

#### 2.3.1. Collection of Plant Volatiles by Headspace-Solid Phase Microextraction (HS-SPME)

Volatile collections focused on phloem from healthy and infected trees using HS-SPME. Tree limbs (length: 1 m; diam.: 7 cm) from healthy and infected trees were obtained, properly packaged to keep them cool, and immediately taken from Ningxia to Beijing. Phloem (10 g) was obtained by cutting 0.5 cm sections. Then, the phloem was placed into the extraction bottle (20 mL). The volatile compounds were collected from extraction bottles using solid phase microextraction (SPME) fiber (50/30 μm DVB/CARBOXEN/PDMS, Supelco, Inc., Bellefonte, PA, USA). The SPME fiber was placed in the inlet of the gas chromatograph and purged at 260 °C for 30 min before each experiment, then the fiber was inserted into the bottle and placed above the materials, extracting for 30 min at 70 °C. After extraction, the fiber was withdrawn from the bottle and inserted into the inlet rapidly. The fiber was remained for 1 min after it was extended, and then desorbed at 260 °C for 5 min. Finally, the fiber was removed for GC-MS (GCMS-QP2010 SE, Shimadzu, Kyoto, Japan) analysis.

#### 2.3.2. Collection of Plant Volatiles by Dynamic Headspace Method

Volatiles emitted from (1) phloem from healthy and infected trees, (2) annual branches, and (3) foliage from seedlings were sampled from the field and collected by dynamic headspace method at the Forestry Bureau in Lingwu city. In the laboratory, each plant part was placed inside the collection bag (Oven Bags Turkey Size, Reynolds Consumer Products, Inc., Lake Forest, IL, USA) and tied tightly. After the air in the bag was drained by the atmospheric sampler (QC-1S, Beijing Municipal Institute of Labour Protection), the filtered air was pumped into the bag through the glass tube with activated charcoal using the atmospheric sampler. The other corner of the bag was attached to a glass tube with 70 mg of Porapak Q (80–100 mesh), the adsorbents were sandwiched between glass wool in the glass tube. The atmospheric sampler pumped clean air at a flow rate of 150 mL/min. Volatiles were collected by the adsorbents for 24 h and were then eluted with 800 μL of n-hexane. The eluent was kept at −20 °C and brought back to Beijing for analysis. Volatiles of empty collection bags were collected as a control.

Volatiles collected by HS-SPME and dynamic headspace method were analyzed by GC-MS-QP2010 SE (Shimadzu, Kyoto, Japan) following the same procedure. The GC was equipped with a Restek Rtx-5MS capillary column (30 m × 0.25 mm × 0.25 μm). Helium was used as the carrier gas at a constant flow rate of 1 mL/min, the injection volume was 1.0 μL in the split mode with a 40:1 split. The oven temperature started at 50 °C, and was increased at 6 °C/min to 180 °C and held for 6 min, then increased at 10 °C/min to 280 °C and held for 10 min. The mass spectra were recorded in the electron impact mode at 70 eV (source at 220 °C, scanned mass range: 29–500 m/z). Data analyses were performed using GCMS solution 4.1.1 (Shimadzu, Japan) with the National Institute of Standards and Technology (NIST) database. The relative content of each volatile compound was identified by peak area calculated by the normalized method [21].

### 2.4. Electroantennogram Responses

The antennal response of both sexes of adult *E. scrobiculatus* and *E. brandti* to the synthetic olfactory stimuli were quantified by electroantennogram (EAG) using the most abundant compounds identified in the previous study. For this experiment we evaluated 1-hexanol (98%; Tianjin Guangfu Fine Chemical Research Institute, Tianjin, China), (1*S*)-(−)-*β*-pinene (98%; Shanghai Macklin Biochemical Co., Ltd., Shanghai, China), (1*R*)-(+)-*α*-pinene (99%; Shanghai Aladdin Bio-Chem Technology Co., Ltd., Shanghai, China), isooctyl alcohol (99%) and liquid paraffin (both were purchased from Tianjin Yongda Chemical Reagent Co., Ltd., Tianjin, China), *cis*-3-hexen-1-ol (98%, Shanghai Yuanye Biotechnology Co., Ltd., Shanghai, China). Each compound was tested singly at five concentrations (100, 10, 1, 0.1, 0.01 μg/μL) using liquid paraffin as solvent. The material was applied (10 μL) to a filter paper strip (5 × 25 mm^2^). Because the weevils showed steady peaks in response to *cis*-3-hexen-1-ol through preliminary test, *cis*-3-hexen-1-ol (30 μg/μL) was selected as a reference compound. The control stimulus was liquid paraffin.

The experiment was carried out as described by Ren [22]. Briefly, weevil antennae were cut off as close as possible to the base of the clavola and took 0.5 mm off the terminal. Then, the antenna was positioned in parallel across a forked metal electrode using Spectra 360 electrode gel (Parker Laboratories Inc., Orange, NJ, USA). The electrode was connected via an interface box (INR-II, Syntech, Hilve rsum, The Netherlands) to a signal acquisition system (IDAC-4, Syntech, Hilve rsum, The Netherlands) connected to a computer using AutoSpike software (Syntech, Hilve rsum, The Netherlands). Both constant airflow and air puffs were generated with a stimulus flow controller (CS-55; Syntech, Hilve rsum, The Netherlands). The controller including a glass tube (2 cm diameter), with an outlet facing towards the antenna at a distance of 1 cm, provided airflow at 30 mL/s. The glass tube presented one lateral hole that permitted the delivery of the stimulus puff inside the tube with the aim of a glass Pasteur pipette attached to Tygon tubes leading to an air source programmed to deliver a 0.5 s pulse at 30 mL/s. The tested odor was carried out by continuous flow of clean air through a Pasteur pipette containing the filter paper strip soaked in the compound. The chemical stimuli were tested randomly from low concentration to high concentration for each compound, with a stimulus duration time of 0.5 s, followed by a 1 min purge with purified air. Liquid paraffin and *cis*-3-hexen-1-ol (30 μg/μL) were successively applied at the beginning and at the end of each compound. EAG responses were measured as the maximum amplitude of depolarization (mV). The response to the solvent control was subtracted from all of the initial responses, and the normalized EAG responses were presented as the ratio of the EAG responses of tested compound to the EAG responses of the standard compound. Each compound was tested on 5 individual male and female antennae. 

### 2.5. Data Analyses

We compared the response of each weevil species to bare phloem of healthy and injured tree by means of two-way Analysis of Variance (ANOVA). One-way ANOVA was used to compare the behavioral responses of *E. scrobiculatus* and *E. brandti* to different parts of *A. altissima*. Whenever appropriate, post-hoc comparisons were done using Duncan’s new multiple range test (*p* < 0.05). For the experiments involving Y-tube olfactometer, one-sample χ^2^ tests were conducted on the numbers of test weevils of each species that made a choice to test the null hypothesis of no preference for a particular synthetic compound vs. clean air. EAG response data for different concentrations of each compound for the same sex were analyzed by one-way ANOVA followed by Duncan’s new multiple range test (*p* < 0.05). We compared EAG responses between female and male *E. scrobiculatus* and *E. brandti* for each treatment using *t*-tests. Parametric data were checked for the assumption of normality and homoscedasticity. All analyses were conducted using IBM SPSS statistics (version: 23; SPSS Inc., Chicago, IL, USA). 

## 3. Results

### 3.1. Attractiveness of Various Parts of A. altissima to Adult E. scrobiculatus and E. brandti

In the first series of experiments, we evaluated the level of attractiveness of bare phloem of healthy and injured *A. altissima* trees to adults of both weevil species. In the field study, we recovered 10% and 14% of the marked *E. scrobiculatus* adults and 37% and 35% of the marked *E. brandti* adults from bare phloem of injured trees and healthy trees, respectively. The response of the weevils differed significantly between species (*F* = 17.07, df = 1, *p* < 0.05), but the state of the tree (i.e., injured vs. healthy) had no effect on the number of weevils responding for each species (*F* = 0.03, df = 1, *p* = 0.865). The interaction term (weevil species × state of tree) was non-significant (*F* = 0.27, df = 1, *p* = 0.613).

In the large-arena experiment, there were significant differences in the level of response of *E. scrobiculatus* and *E. brandti* to the various parts of *A. altissima*. Adults of *E. scrobiculatus* significantly preferred phloem from the trunk (*F* = 12.57, df = 3, 31, *p* < 0.05). *E. brandti* adults responded in significantly greater numbers to the tree limb than to any other plant material. Phloem from trunk ranked second in preference, and this material was more attractive to the weevils than annual branches and seedling foliage (*F* = 13.36, df = 3, 27, *p* < 0.05) (Figure 1).

Results from the Y-tube olfactometer revealed that *E. scrobiculatus* females responded positively to all types of plant material that they were exposed to when compared with air (healthy phloem versus air: χ^2^ = 11.267, df = 1, *p* < 0.01; injured phloem versus air: χ^2^ = 10.889, df = 1, *p* < 0.01; annual branches versus air: χ^2^ = 9, df = 1, *p* < 0.01). Similar results were found for *E. scrobiculatus* males (injured phloem versus air: χ^2^ = 12.25, df = 1, *p* < 0.01; seedling foliage versus air: χ^2^ = 12.25, df = 1, *p* < 0.01; annual branches versus air: χ^2^ = 12.25, df = 1, *p* < 0.01) (Figure 2). 

Contrasting gender responses were recorded for *E. brandti*. Female and male weevils showed a significant preference for phloem regardless of the condition of the tree (healthy phloem: female: χ^2^ = 12.25; *p* < 0.01; males: 100% of the males responded to healthy phloem; injured phloem: female: χ^2^ = 14.22; *p* < 0.01; male: χ^2^ = 4; *p* < 0.05) when compared with air. However, a repellent effect was noted for females, which significantly selected the arm associated with clean air compared with the arm containing foliage from seedlings (χ^2^ = 5.33; *p* < 0.05) whereas males showed a significant preference for seedling foliage. While no significant differences were observed in the response of females toward volatiles emitted from annual branches (χ^2^ = 2.273, *p* > 0.05), it exerted a significant repellent effect on males (χ^2^ = 6.25; *p* > 0.05) (Figure 3). 

### 3.2. Characterization of Volatiles from Different Parts of A. altissima

The volatiles from phloem from healthy and infested trees were collected by HS-SPME. Twenty-three and thirty-five components were identified in the volatiles of phloem from healthy and injured tree, respectively. There were more types of compounds in the volatiles of phloem from injured trees than from healthy trees (Table 1).

A total of 43 compounds were identified from different parts of *A. altissima* by dynamic headspace method. Fourteen volatile compounds were identified from phloem from healthy tree, and the main components of volatiles included *β*-caryophyllene (33.88%), 2-phenylethyl-1,1,2,2-d4-amine (17.52%), (−)-camphene (8.22%) and 1-tridecanol (6.81%), accounting for 66.43% of the total amount. Twenty-two components were identified in the volatiles of phloem from injured tree, and the primary compounds were 1-tetradecene (17.58%), 1-tridecene (10.05%), *β*-copaene (8.94%), and n-hendecane (6.91%), accounting for 43.48% of the total amount. Eleven volatile compounds were identified from seedling foliage, and the main compounds included *β*-copaene (41.72%), *β*-caryophyllene (26.94%), *α*-farnesene (12.74%), and leaf acetate (11.7%), accounting for 93.1% of the total amount. Eight volatile compounds were identified from annual branches, and the main compounds included *β*-copaene (41.43%), *α*-farnesene (22.29%), *β*-caryophyllene (21.63%), and *β*-elemene (8.24%), accounting for 93.59% of the total amount (Table 2).

The number of volatile components from phloem of injured trees exceeded that from phloem of healthy tree as determined by the HS-SPME and dynamic headspace methods. The main components were alkenes (76.04%) and alcohols (56.69%) in phloem of injured and healthy tree by HS-SPME, respectively. The main volatile components of different parts of *A. altissima* by dynamic headspace method were alkenes, especially for seedling foliage (84.37%) and annual branches (94.33%). Besides, alkenes accounted for 53.05% and 50.17% of the total volatile compounds in phloem of healthy and injured trees, respectively (Table 1 and Table 2).

### 3.3. Electroantennogram Responses

We analyzed EAG responses of female and male of *E. scrobiculatus* and *E. brandti* to five selected volatile compounds emitted by *A. altissima* phloem. The EAG responses to each compound differed significantly between female and male *E. scrobiculatus* and *E. brandti*. In particular, female responses to 1-hexanol were significantly greater than those of males at moderate doses (e.g., at 0.1, 1, and 10 μg/μL) in *E. scrobiculatus* and *E. brandti*. Male responses to (1*S*)-(−)-*β*-pinene were significantly greater than those of females at higher doses (e.g., at 0.1, 1, 10, and 100 μg/μL) in *E. brandti* (Table 3, Figure 4 and Figure 5).

## 4. Discussion

Using a comparative approach, we investigated the role of host plant-derived volatiles in the foraging behavior of adults of *E. scrobiculatus* and *E. brandti* via behavioral and antennal responses of both weevil species. Under field conditions, *E. brandti* adults feed on the trunk. In the field study, we found many weevils aggregated on the bare phloem of healthy and injured trees, while they preferred to feed on the tree limb of *A. altissima* in the large-arena experiment. In the Y-tube bioassay, female and male *E. brandti* significantly preferred to volatiles from phloem, regardless of tree condition. Under field conditions, *E. scrobiculatus*, adults feed on 1-year-old branches, perennial branches, and petioles. In the field study, we found that weevils aggregated on the bare phloem of healthy and injured trees. Adult *E. scrobiculatus* significantly preferred phloem volatiles over trunk volatiles in the large-arena experiment whereas in the Y-tube bioassay female and male weevils responded positively to all types of plant parts that they were exposed to. Hence, while both weevil species have the ability to discriminate against different plant parts, *E. scrobiculatus* responses to host plant material seem to be more variable compared with *E. brandti*. Overall, our findings indicate that volatile compounds of *A. altissima* may act as important olfactory cues in feeding behavior, particularly the volatiles from phloem for *E. brandti*.

Volatiles of different parts of *A. altissima* were collected by two methods (HS-SPME and dynamic headspace). It is known that plants may release more quantities of volatile chemicals when damaged by herbivorous insects [23]. In this study, the number of volatile compounds of phloem from weevil-damaged trees were greater than those identified from healthy trees, regardless of the collection method (HS-SPME or dynamic headspace method). When compared to volatiles from phloem, there were fewer volatile compounds identified in annual branches and seedling foliage. While volatile compounds emitted by *A. altissima* have been analyzed before, there has been some variability in results. For example, Mastelić and Jerković [24] analyzed the chemical composition of volatile compounds from fresh and dried leaves of young and old trees, and the main constituents were aliphatic C_6_-compounds and sesquiterpenes. Volatile compounds from leaves were extracted by simultaneous distillation extractor (SDE) and SPME, the component with the highest relative content was caryophyllene, and there were more compounds being detected in SPME than SDE extracts [25]. In turn, Xie [26] analyzed the volatile components of branches using the dynamic headspace method and found that *α*-pinene was the most abundant component. Ji et al. [27] analyzed the volatile components of leaves by HS-SPME, and the main components were (4*E*)-4-hexenyl acetate and (*Z*)-hex-3-en-1-ol. In this study, the main component in phloem from healthy and injured trees by HS-SPME was 1-hexanol and *β*-caryophyllene, respectively. The main component in phloem from healthy and injured trees, seedling foliage and annual branches by dynamic headspace method was *β*-caryophyllene, 1-tetradecene, *β*-copaene, and *β*-copaene, respectively. Overall, the type of collection method seems to have a strong influence on the qualitative and quantitative composition of *A. altissima* volatile compounds. In order to more precisely identify the volatile components of different parts of *A. altissima*, the methods of collection and analysis of volatiles need to be further refined. 

This investigation provided the first evidence that *E. scrobiculatus* and *E. brandti* respond positively to volatiles emitted by their host plant. Females and males of *E. scrobiculatus* and *E. brandti* showed a significant preference for healthy phloem over clean air in the Y-tube olfactometer. Consequently, five compounds that were relatively more abundant in healthy phloem as determined by HS-SPME were selected as stimuli to test EAG responses of weevils. Due to the limitation of weevils, we did not test more individual compounds or blends. Although both weevil species could be attracted by volatiles from healthy phloem of *A. altissima*, there were different responses to tested compounds between them. Chemosensory genes were identified in *E. scrobiculatus* and *E. brandti* by antennal transcriptome sequencing, and the odorant binding proteins of *E. scrobiculatus* and *E. brandti* show different expression patterns [28]. It is plausible that each species shows different abilities to recognize volatiles that are involved in feeding and reproduction. Other studies have shown that insect behavioral responses to host volatile blends can exceed the responses to individual components [20]. Usually, blends of compounds are needed to elicit adequate behavioral responses by a foraging insect [29,30,31]. Consequently, we need to explore the response of weevils to host volatile blends in the future.

The morphological characteristics and physical conditions of plants may influence the feeding behavior of phytophagous insects. Dimock and Tingey [32] studied the effect of potato glandular trichomes on host acceptance behavior of Colorado potato beetle (*Leptinotarsa decemlineata*) larvae. They found that removal of the trichome barrier by wiping leaflets with tissue paper lead to increased incidence of feeding by larvae. Elkinton and Wood [33] studied the feeding and boring behaviors of *Ips paraconfusus* Lanier on the bark of a host (pine) and non-host tree (fir) species, and they found that the beetle preferred the pine to fir phloem when the outer bark was removed and the phloem was retained intact. However, no preferences were apparent for the intact pine or fir outer bark when the phloem was removed. 

Under natural conditions, *E. scrobiculatus* and *E. brandti* often feed on the tree-of-heaven together, so releasing them together in the large-arena experiment was meant to simulate the field condition. *E. scrobiculatus* adults feed on annual branches, perennial branches and petioles in the field [12]. Here, we found that a few weevils responded to volatiles emitted by bare phloem of healthy and injured trees, and adults significantly preferred phloem from the trunk in large-arena experiment, whereas female and male weevils were significantly attracted by all the tested materials in the Y-tube bioassay. This suggests that the responses observed in the field may include factors other than plant volatiles. In the case of *E. brandti*, the observed behavior of adults in the field matched more closely the olfactory-based responses documented in this study.

## 5. Conclusions

Our findings provided an insight into the potential role that volatiles emitted by *A. altissima* played in the feeding behavior of *E. scrobiculatus* and *E. brandti*. This study demonstrated that (1) volatile compounds of *A. altissima* may act as important olfactory cues in feeding behavior, particularly the volatiles from phloem for *E. brandti,* and (2) some similarities and some differences exist in the way host–plant-derived volatiles influence the behavior of *E. scrobiculatus* and *E. brandti*. We confirmed the hypothesis that plant volatiles play an important role in the foraging behavior of *E. brandti* and *E. scrobiculatus*. Adult *E. brandti* consistently preferred volatiles from phloem of *A. altissima*, so it is vital to further analyze the responses of *E. brandti* to volatile compounds emitted by phloem. This will be helpful for developing effective plant-based attractants to monitor and potentially develop attract-and-kill systems for *E. brandti*.

## Figures and Tables

**Figure 1 insects-12-00068-f001:**
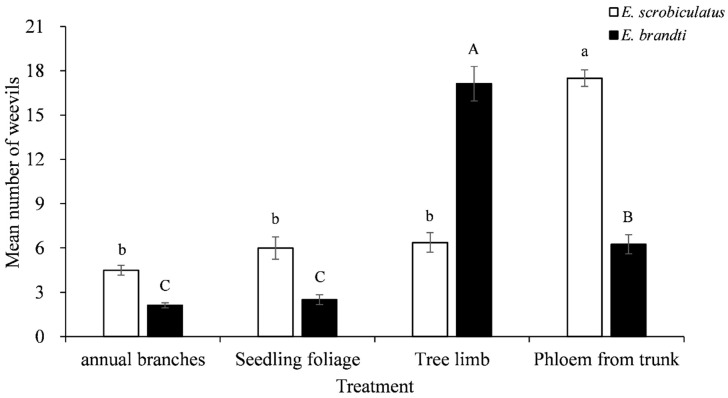
Response of adult *E. scrobiculatus* and *E. brandti* to various types of plant material from *A. altissima* in the large-arena experiment. Means (±SE) were calculated from the number of weevils responding to each treatment. For each weevil species, different letters above bars denote significant differences according to ANOVA and Duncan’s new multiple range test at *p* = 0.05.

**Figure 2 insects-12-00068-f002:**
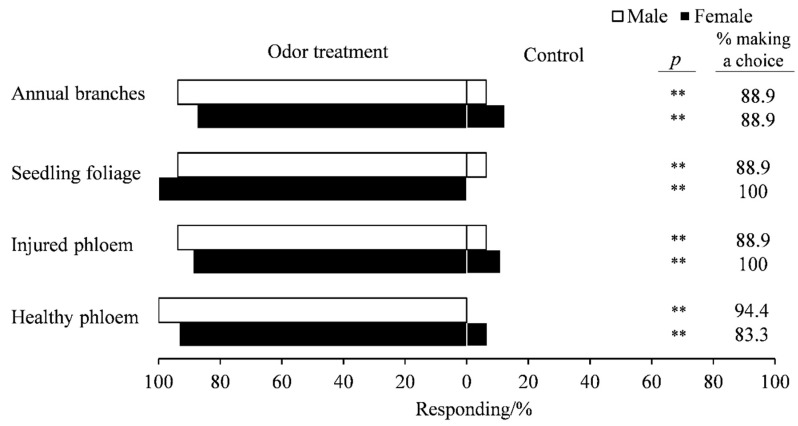
Response of *E. scrobiculatus* females (*n* = 18) and males (*n* = 18) to different parts of *A. altissima* in the Y-tube olfactometer. Injured phloem = phloem from injured trees; healthy phloem = phloem from healthy trees. *p*-values are based on Pearson’s chi-square test: **, *p* < 0.01.

**Figure 3 insects-12-00068-f003:**
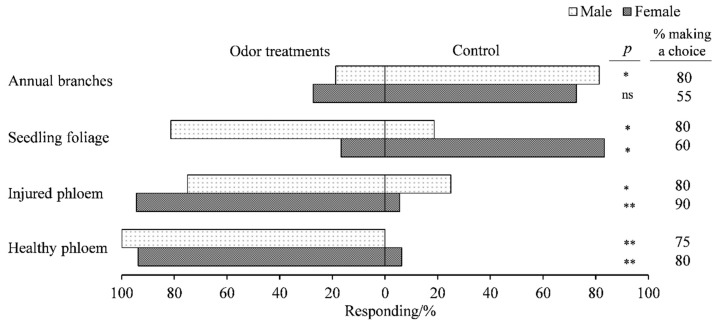
Response of *E. brandti* females (*n* = 20) and males (*n* = 20) to different parts of *A. altissima* in the Y-tube olfactometer. Injured phloem = phloem from injured trees; healthy phloem = phloem from healthy trees. *p*-values are based on Pearson’s chi-square test: *, *p* < 0.05; **, *p* < 0.01; ns, *p* ≥ 0.05.

**Figure 4 insects-12-00068-f004:**
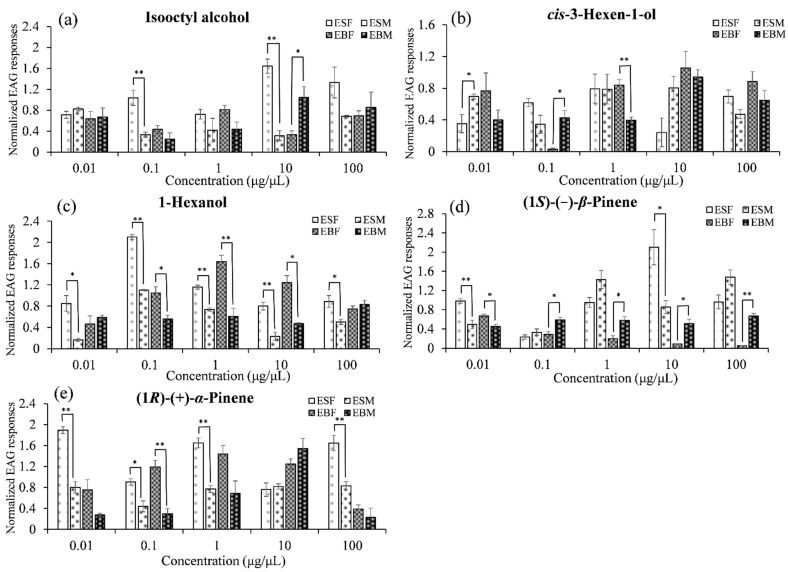
Mean EAG responses (± SEM) of female *E. scrobiculatus* (=ESF), male *E. scrobiculatus* (=ESM), female *E. brandti* (=EBF) and male *E. brandti* (=EBM) to five compounds at five dosages (0.01, 0.1, 1, 10 and 100 μg/μL). (**a**): Isooctyl alcohol; (**b**): *cis*-3-Hexen-1-ol; (**c**): 1-Hexanol; (**d**): (1*S*)-(−)-*β*-Pinene; (**e**): (1*R*)-(+)-*α*-Pinene. Asterisks indicate significant differences (* = *p* < 0.05; ** = *p* < 0.01).

**Figure 5 insects-12-00068-f005:**
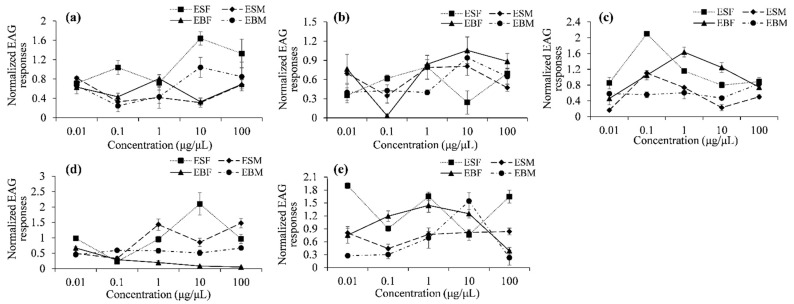
Dose-dependent EAG responses of female *E. scrobiculatus* (=ESF), male *E. scrobiculatus* (=ESM), female *E. brandti* (=EBF) and male *E. brandti* (=EBM) to five compounds. (**a**): Isooctyl alcohol; (**b**): *cis*-3-Hexen-1-ol; (**c**): 1-Hexanol; (**d**): (1*S*)-(−)-*β*-Pinene; (**e**): (1*R*)-(+)-*α*-Pinene.

**Table 1 insects-12-00068-t001:** Volatile compounds of healthy and injured phloem of *A. altissima* by headspace solid phase microextraction (HS-SPME).

Number	Name	CAS	Area under Peak	
			Healthy Phloem	Injured Phloem
1	(1*R*)-(+)-*α*-Pinene	7785-70-8	599,387	302,479
2	Camphene	79-92-5	289,369	
3	*β*-Phellandrene	555-10-2	78,859	
4	(1*S*)-(−)-*β*-Pinene	18172-67-3	358,576	
5	Myrcene	123-35-3		1,360,591
6	(+)-(4*R*)-Limonene	5989-27-5		1,898,791
7	*β*-Ocimene	3338-55-4		4,851,574
8	(+)-2-Carene	554-61-0		1,210,188
9	*α*-Cubebene	17699-14-8		12,832,275
10	Copaene	3856-25-5		10,567,644
11	*β*-Caryophyllene	87-44-5	65,276	42,295,799
12	*α*-Humulene	6753-98-6		7,472,179
13	*β*-Copaene			2,172,542
14	(+)-*δ*-Cadinene	483-76-1		1,005,043
15	l-Calamenene	483-77-2		288,668
16	Caryophyllene oxide	1139-30-6		616,535
17	2-Xylene	95-47-6	184,083	1,196,424
18	*p*-Xylene	106-42-3	332,852	
19	1-Ethyl-4-methylbenzene	622-96-8	68,356	
20	3-Ethyltoluene	620-14-4		972,029
21	2-Ethyltoluene	611-14-3		560,994
22	Mesitylen	108-67-8	75,378	5,228,481
23	1,2,3-Trimethylbenzene	526-73-8	69,816	566,908
24	1-Methyl-3-propylbenzene	1074-43-7		445,924
25	2-Ethyl-p-xylene	1758-88-9		981,883
26	4-Ethyl-m-xylene	874-41-9		785,844
27	1,2,3,4-Tetramethylbenzene	488-23-3		520,840
28	1,2,3,5-Tetramethylbenzene	527-53-7		864,315
29	1,2,4,5-Tetramethylbenzene	95-93-2		303,672
30	3-Methylcyclopentanol	18729-48-1	159,700	
31	*cis*-3-Hexen-1-ol	928-96-1	863,559	
32	Cyclohexanol	108-93-0	333,474	
33	1-Hexanol	111-27-3	4,902,291	7,042,158
34	2-Ethyl-1-hexanol	104-76-7	515,699	
35	3,3-Dimethyl-1,2-epoxybutane	2245-30-9	1,169,014	
36	2,4-Dimethylhexane	589-43-5	110,544	
37	Tetradecane	629-59-4	57,964	
38	*d*-Camphor	464-49-3	59,699	
39	2-Hendecanone	112-12-9		331,169
40	Tetradecanal	124-25-4		644,070
41	Pentadecanal	2765-11-9		286,561
42	Dibutyl ether	142-96-1	1,178,162	1,392,083
43	1,2-Dimethoxybenzene	91-16-7		540,650
44	Dibutyl phthalate	84-74-2	272,435	554,530
45	Oxetane, 3-(1-methylethyl)-	10317-17-6	85,020	
46	Di-tert-butyl peroxide	110-05-4	119,142	
47	4-Hydroxy-3-methylbutanal	56805-34-6		972,544
48	Aciphyllene	87745-31-1		330,730
49	Cubebene	13744-15-5		2,455,168
50	(+)-Epi-bicyclosesquiphellandrene	54274-73-6		367,917

**Table 2 insects-12-00068-t002:** Volatile compounds of different parts of *A. altissima* by dynamic headspace method.

Number	Name	CAS	Relative Content (in %)			
			Healthy Phloem	Injured Phloem	Seedling Foliage	Annual Branches
1	(1*R*)-(+)-*α*-Pinene	7785-70-8	2.92		0.53	
2	(−)-Camphene	5794-04-7	8.22			
3	Camphene	79-92-5			0.97	
4	*β*-Pinene	127-91-3	5.49			
5	(1*S*)-(−)-*β*-Pinene	18172-67-3			0.54	
6	1-Decene	872-05-9		1.7		
7	(+)-(4*R*)-Limonene	5989-27-5		1.21		
8	1-Undecene	821-95-4		2.01		
9	1-Tridecene	2437-56-1		10.05		
10	*α*-Pinene	3856-25-5		1.44	0.93	0.74
11	*β*-Bourbonene	5208-59-3		4.33		
12	1-Tetradecene	1120-36-1		17.58		
13	*β*-Elemene	515-13-9				8.24
14	*β*-Caryophyllene	87-44-5	33.88		26.94	21.63
15	1-Pentadecene	13360-61-7	2.54			
16	*β*-Copaene			8.94	41.72	41.43
17	*α*-Farnesene	502-61-4			12.74	22.29
18	1-Heptadecene	6765-39-5		2.91		
19	2,4-Dimethylheptane	2213-23-2		1.62		
20	n-Hendecane	1120-21-4		6.91		
21	Dodecane	112-40-3		1.2		
22	n-Tridecane	629-50-5	2.88	5.45		
23	Tetradecane	629-59-4	3.34	2.67		
24	n-Pentadecane	629-62-9	3.13	6.01		
25	n-Heptadecane	629-78-7		1.12		
26	1-Dodecanol	112-53-8		11		
27	1-Tridecanol	112-70-9	6.81			
28	1-Pentadecanol	629-76-5		1.21		
29	Ethyl 2-methylbutyrate	7452-79-1			2.24	
30	Ethyl tiglate	5837-78-5			0.67	
31	Leaf acetate	3681-71-8			11.7	1.65
32	Bis(2-ethylhexyl) adipate	103-23-1				0.63
33	Carvacrol	499-75-2				3.39
34	2,4-Di-tert-butylphenol	96-76-4	2.39			
35	2-Phenylethyl-1,1,2,2-d4-amine	876-20-0	17.52			
36	Cuminaldehyde	122-03-2	6.1			
37	4-Ethylbenzaldehyde	4748-78-1	2.15			
38	Phenylethylene	100-42-5		6.12		
39	Isobutylbenzene				1.04	
40	Hexane,2,2,3,3-tetramethyl-	13475-81-5		1.59		
41	2-Pentylfuran	3777-69-3		2.82		
42	4-Ethylcumen	4218-48-8		2.11		
43	5-Ethylundecane	17453-94-0	2.63			

**Table 3 insects-12-00068-t003:** Dose-dependent electroantennography (EAG) responses of *E. scrobiculatus* and *E. brandti* to different compounds. Means (±SE) followed by the same letter in the same line are not significantly different (*p* < 0.05).

Compound	Insect	Mean EAG Responses (± SE)				
		0.01 μg/μL	0.1 μg/μL	1 μg/μL	10 μg/μL	100 μg/μL
Isooctyl alcohol	ESF	0.71 ± 0.07c	1.04 ± 0.14bc	0.72 ± 0.1c	1.64 ± 0.14a	1.33 ± 0.29ab
	ESM	0.82 ± 0.04a	0.33 ± 0.05b	0.42 ± 0.22b	0.31 ± 0.1b	0.68 ± 0.02ab
	EBF	0.63 ± 0.14abc	0.43 ± 0.08bc	0.81 ± 0.08a	0.33 ± 0.08c	0.69 ± 0.09ab
	EBM	0.67 ± 0.17ab	0.25 ± 0.12b	0.43 ± 0.14ab	1.04 ± 0.21a	0.85 ± 0.3ab
*cis*-3-Hexen-1-ol	ESF	0.35 ± 0.12ab	0.62 ± 0.05ab	0.79 ± 0.18a	0.24 ± 0.18b	0.7 ± 0.08a
	ESM	0.7 ± 0.03ab	0.34 ± 0.11b	0.79 ± 0.19a	0.81 ± 0.14a	0.47 ± 0.06ab
	EBF	0.77 ± 0.22a	0.03 ± 0.01b	0.84 ± 0.07a	1.06 ± 0.21a	0.88 ± 0.12a
	EBM	0.4 ± 0.13b	0.43 ± 0.09b	0.4 ± 0.04b	0.94 ± 0.09a	0.65 ± 0.12ab
1-Hexanol	ESF	0.85 ± 0.15c	2.1 ± 0.05a	1.16 ± 0.04b	0.8 ± 0.07c	0.89 ± 0.11bc
	ESM	0.16 ± 0.03d	1.1 ± 0.01a	0.74 ± 0.03b	0.23 ± 0.08d	0.5 ± 0.05c
	EBF	0.46 ± 0.16d	1.05 ± 0.11bc	1.64 ± 0.12a	1.25 ± 0.13b	0.75 ± 0.05cd
	EBM	0.58 ± 0.04ab	0.56 ± 0.07ab	0.61 ± 0.16ab	0.47 ± 0.01b	0.83 ± 0.08a
(1*S*)-(−)-*β*-Pinene	ESF	0.98 ± 0.05b	0.23 ± 0.05c	0.95 ± 0.1b	2.1 ± 0.36a	0.96 ± 0.15b
	ESM	0.49 ± 0.09bc	0.33 ± 0.07c	1.43 ± 0.19a	0.85 ± 0.14b	1.48 ± 0.15a
	EBF	0.67 ± 0.04a	0.29 ± 0.05b	0.2 ± 0.06bc	0.08 ± 0.0048cd	0.05 ± 0.0043d
	EBM	0.45 ± 0.05a	0.59 ± 0.05a	0.58 ± 0.07a	0.51 ± 0.09a	0.67 ± 0.05a
(1*R*)-(+)-*α*-Pinene	ESF	1.9 ± 0.07a	0.91 ± 0.06b	1.65 ± 0.1a	0.76 ± 0.13b	1.65 ± 0.15a
	ESM	0.8 ± 0.11a	0.44 ± 0.1b	0.77 ± 0.06b	0.82 ± 0.05b	0.83 ± 0.08b
	EBF	0.75 ± 0.19b	1.19 ± 0.12a	1.44 ± 0.16a	1.25 ± 0.09a	0.39 ± 0.08b
	EBM	0.28 ± 0.03b	0.3 ± 0.1b	0.69 ± 0.24b	1.54 ± 0.19a	0.23 ± 0.18b

## Data Availability

Data sharing not applicable.

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
