# Peer review of "Contrasting Behavioral and Electrophysiological Responses of Eucryptorrhynchus scrobiculatus and E. brandti (Coleoptera: Curculionidae) to Volatiles Emitted from the Tree of Heaven, Ailanthus altissima"

_insects, 2021, doi:10.3390/insects12010068_

Round 1
Reviewer 1 Report
General comments:
Collection of plant volatiles by dynamic headspace method.
Authors did not indicate how many replicates were done. As there is no SE given in the table 2, one can assume that there are no replicates available which in turn don’t allow statistical data evaluation comparing amounts of volatiles trapped from different sample types.
Determination of relative quantities of VOCs by SPME DOES NOT ALLOW one to compare quantities of different compounds in the same sample. Please, see a link explaining methodological requirements using SPME. https://static.springer.com/sgw/documents/874338/application/pdf/SPME%20Guidelines.pdf
Moreover, no internal standards were used to monitor saturation of fiber.
Specific comments
Line 102-103. Why information about the two choice experiments is presented before multiple choice experiments. Were both types of experiments carried out at Forestry Bureau in Lingwu city? If so, please remove “Two choice”.
Lines 105-108. Annual branches, foliage, and tree limb are not host-plant tissues. They are certain parts of a plant.
Line 132. What was a diameter of the Y-tube olfactometer?
Lines 152-154. The identity and quantities of host-plant volatiles can’t be determine using only HS-SPME and dynamic headspace methods. These two methods are used for sampling of volatiles but not for identification.
Lines 161-162. “The SPME fiber was placed in the inlet of the gas chromatograph and purged at 260°C for 30 min before each experiment,…”
Just comment. No change needed. Your SPME fiber purification time and temperature was almost the same as conditioning. Conditioning procedure is advised to carry out once for a new fiber. There is no need to purify fiber so long unless you have checked that certain volatile compounds were not desorbed from a fiber. Long expose to high temperatures shorten fiber life time and it starts to degrade quicker.
Lines 163-165. “After extraction, the fiber was withdrawn from the bottle and inserted into the inlet rapidly. The fiber was extended for 1 min, and then desorbed at 260°C for 5 min.”
I don’t understand how desorbtion could las for 5 min if fiber was extended just for 1 min.
Lines 180-185. The description of GC-MS procedure is incomplete and it would not be possible to reproduce the analytical procedure based on the description. For example at which mode GC injector was operated, what was caring gas, what was gas flow through analytical column, what type of a capillary column was used and etc.
Lines 198-207. The description of EAG set up and procedure is not clear. Authors wrote “…the tested odor was carried out by continuous flow of clean air through a Pasteur pipette containing the compound. … a stimulus duration time of 0.5 s,…”.
Lines 319-321. “Hence, we infer that E. scrobiculatus have the ability to discriminate against different types of tissue depending on the environmental conditions.”
I don’t see any experiment taking into consideration environmental conditions. Authors have to define what they mean by environmental conditions.
Lines 254-257. “However, a repellent effect was noted for females, which significantly selected the arm associated with air compared with the arm containing foliage from seedlings (χ2 = 5.33; p < 0.05) whereas all males responded to foliage from seedlings over air.”
Your statement regarding male response is not clear. What did you mean by “all males responded”? Based on the presentation of the females’ data I would consider male response as male choice. In that case, your statement contradicts the figure 3 where around 75% but not all males choose seedling over air sample.
Technical comments
Lines 66-70. The sentences are grammatically incorrect.
Line 81. Change electroantennogram to electroantennograph.
Line 88. Remove the full stop and adjust the sentence which was started with regular character after full stop.
Line 95. Please, correct 0600 h to 06.00 h Make the corrections to time format in the rest of the text as well.
Line 136. Please change 250 ml/min to 250 mL/min.
Lines 160-161 “…DVB/CARBOXEN-PDMS…” use the same way to represent the mixture of three absorbents, i.e. either DVB/CARBOXEN/PDMS or DVB-CARBOXEN-PDMS
Lines 165-166. “Finally, the fiber was removed for GC-MS (GCMS-QP2010 SE, 165 Shimadzu, Japan) analysis.”
Please give a full name of the GC-MS truncation and provide model of gas chromatograph and mass spectrometer. QP2010 is the model of MS as I understand.
Line 187. E. scrobiculatus and E. brandti has to be italic.
Reviewer 2 Report
The paper by Wen et al. proposes an interesting approach to explore the role of host plant-derived volatiles in feeding selection behaviour of the two weevils, Eucryptorrhynchus scrobiculatus and E. brandti (Coleoptera: Curculionidae). The text is easily readable and quality of presentation is in the average but, in my opinion, some relevant concerns must be solved to allow the paper to be considered for publication in Insects. In detail:
- The EAG part of the research is out of scientific soundness. There is no reason to extract volatiles from healthy and injured plants, and in different tissues, using two methods of extraction, despite the literature available about volatiles emitted by Ailanthus altissima , and to test by EAG only 5 compounds.
I know that in line 351-355 the authors state that “Females and males of E. scrobiculatus and E. brandti showed a strong bias toward healthy phloem, so five compounds that had a higher relative content from healthy phloem by HS-SPME were selected as stimuli to test EAG responses of weevils” but I found this is not acceptable and it is partially true because I do not see this bias in behavioural data.
Also, I understand the authors when they affirm that “Due to the limitation of experimental conditions, we did not test more individual compounds or blends” but to me, this is not acceptable. it is not possible to perform EAG recordings from only 5 insects for sex and species, they must be at least 10 or 15 to be considered representatives. Single-cell recordings are different because they provide direct evidence of the responses but EAG recordings provide general indirect evidence of the response that must be sustained by a higher number of replicas to be robust. I consider redundant to perform dose-response for all the tested chemicals but I strongly suggest to test more chemicals and more insects.
The last but not the least criticism is about figure 4: it is not clear, the authors have to report the significance of the responses to respect to the control (paraffin oil) and present comparison between species and sexes in a clearer way. Also, dose-response experiments are not analysed, such as performing a curve of the response.
- The behavioural experiments have better scientific soundness but they also present some relevant bias that must be solved:
- a) in the Y-tube olfactometer was not performed choice test between different plant tissues, all the tests were performed vs the empty jar, representing the control;
- b) in the large-arena experiment were used the tree limbs (significantly preferred by brandti) but they were not tested in the olfactometer, resulting in a bias, particularly for E. brandti.
- c) In the discussion the authors do not consider possible effects from chemicals emitted by conspecifics in the large-arena experiment, where more weevils were released together during the bioassay.
- In the Discussion the authors perform some considerations not completely supported by the data reported in the results, e.g.:
Line 325: “In the Y-tube bioassay, female and male E. brandti were significantly attracted by volatiles from the phloem, regardless of tree condition. Overall, our findings indicate that volatile compounds from the phloem of A. altissima may act as important olfactory cues in host plant selection, particularly in the case of E. brandti”.
In olfactometers the authors did not perform choice test so they cannot use simple test vs the control as a choice test, this is not correct! Accordingly, also conclusions must be revised.
Line 352: “Females and males of E. scrobiculatus and E. brandti showed a strong bias toward healthy phloem…”
I do not understand this sentence, no behavioural results in the paper support this consideration.
Considering all the above-reported concerns, the paper must be strongly revised, also performing additional behavioural (choice tests strongly suggested) and electrophysiological (more replicas mandatory, more chemicals strongly suggested) experiments, to be considered for publication in Insects.
Reviewer 3 Report
The manuscript by Wen et al. is an interesting read and shows good potential to serve as a platform for pest management of two weevil species. There are a few points below that need to be addressed.
- The abstract needs improving to better set the rationale for the work undertaken.
- lines 45-47: I do not think this is the right way round. An insect`s ability/potential to exploit a food source is inherent.
- line 88: What do you mean by `conventional`?
- line 96: Do you mean `phloem`?
- line 144: `common arm` - do you mean `stem`?
- lines 179-185: specify column type, carrier gas etc.
- lines 198-207: Provide more details for EAG experiments beyond ref. 22, e.g. hardware/software used, details of antennal preparation etc.
- line 234: You cannot say it is attractiveness, only arrestment/preference. Attraction is a long/medium-distance response, whereas in the arena experiment, test insects had physical access to stimuli. Thus, they could have randomly wandered around and when coming across a preferred stimulus, they were arrested by it.
- Fig. 1: Choose different patterning for species bars, because it is not clear.
- line 268: use `abundance` instead of `content`
- Table 1: I suggest to leave out the `type` column, because some compound classifications are wrong, and also, the differentiation is not always easy. I think this table can stand on its own without this.
- lines 298-303: Why are there no statistical comparisons between species, as outlined in the Materials & Methods section?
- line 322: It cannot be said that volatiles are responsible for insect field response based on this experiment, only that a range of cues guided beetles to exposed phloem. Even in this case, attraction can only be declared if detailed observations were undertaken, showing a directed movement towards phloem.
- lines 329-330: Plants can release more volatiles upon herbivory, but not always. I suggest sentence to be changed accordingly.
- line 330: Not quantities, but number of compounds, because total VOC release has not been determined.
- lines 361-362: "...insect behavioral responses to host volatile blends can exceed the responses..."
- lines 369-372: Please rephrase sentence, because it is not clear.
- Conclusions: I suggest to find a lab with GC-EAD to pick out biologically relevant compounds from phloem extracts, as opposed to using most abundant compounds in blends.
Round 2
Reviewer 1 Report
Comments to the manuscript ID: insects-1031063_version 2.
Title: Contrasting behavioral and electrophysiological 2 responses of Eucryptorrhynchus scrobiculatus and E. brandti (Coleoptera: Curculionidae) to volatiles emitted from the tree of heaven, Ailanthus altissima.
Authors: Xiaojian Wen, Kailang Yang, Jaime C. Piñero, Junbao Wen
General comments:
Table 1. Authors presented amounts as per cent which means comparison of quantities of different compounds in the same sample. Unfortunately, authors did not address my comment that determination of relative quantities of VOCs by SPME DOES NOT ALLOW one to compare quantities of different compounds in the same sample. Please see a link explaining methodological requirements using SPME. https://static.springer.com/sgw/documents/874338/application/pdf/SPME%20Guidelines.pdf
In the table 1, please, present absolute amounts (for example areas under peaks) instead of per centages.
Please use L but not l to indicate liter. See lines 308, 309 Table 3, Fig.4 and some other places.
Please use the same style indicating significance level. In some occasions regular p in another occasions p in italic are used.
Tables 1, 3 and some other places in the text
I suggest to start name of compounds by capital letter. For example, cis-3-Hexen-1-ol instead of cis-3-hexen-1-ol.
Please use UPAC rules to present chemical names properly. For example “cis-“ has to be italic, (1S)- “S” has to be italic as well as “R” in (1R)- has to be italic, in p-xylene p has to be italic.
Specific comments
Line 35. As a rule, complete name of a term followed by truncation is given at the site the first time it appears in a text. As you use EAG once in the abstract, please change “EAG” to “electroantennographic”.
Line 176. Please change “(0.25 mm × 30 m × 0.25 μm)” to “(30 m × 0.25 mm × 0.25 μm)”
Line 177. “…in the split mode with a 40:1 split”. No change is needed, it is just a comment. When SPME is used, please set up injector at the splitless mode. Using split at the ratio 40:1, 39 parts of your sample were lost, i.e. blown away from an injector.
Reviewer 2 Report
I'm really sorry, I understand how difficult could be satisfy my requests, but I really think that the paper in this revised version is not significantly improved in consideration of my main concerns, for this reason, I consider the quality of this paper not acceptable, to be published in Insects. It can be probably adapt to be published in a lower I.F. Journal.
Best regards, Merry Christmas and happy new year
Author Response
Dear reviewer,
I am so sorry that the revised version was not satisfied with you. We read again every comment made by you in the first round, and we also read carefully our replies and the entire manuscript, our replies were honest, detailed, and complete. We really could not test more and more volatiles now, but we indicated in the manuscript that future work would test more volatiles.
We sincerely hope that you will reconsider your decision.
Sincerely,
Xiaojian Wen
Round 3
Reviewer 1 Report
Dear authors,
You have done good job improving the manuscript. I have no more comments and wish you all the best.
Reviewer 2 Report
I'm really sorry,
I read again the revised manuscript very carefully and the authors are right, their replies are honest and complete.
Any case, they did not solve the main problems I pointed in my revision, e.g.:
-behavioural assays should be better designed (in the Y-tube olfactometer all odour stimuli were tested vs the control, in the behavioural experiments insects were never tested individually, olfactory stimuli used in the 3 types of behavioural experiments are slightly different and this affect negatively on the clarity of the results).
-EAG performed on 5 insects cannot be considered reliable in my opinion. The authors state that more test will be performed in the future and that no more recordings have been possible because of the low availability of insect. But this is in contrast with the number of 480 insects for each species collected to perform only 8 replicas of large still-air arena experiments.
- in addition, figs 4 and 5 are not easily readable and the discussion/conclusions soundness must be improved in relation to the aim of the paper and the results.
For these reasons, I'm sorry but as in my last revision, I consider the quality of this paper not acceptable to be published in Insects. I suggest publishing it in a lower I.F. Journal.
Best regards